# Modelling Regime Changes of Dunes to Upper-Stage Plane Bed in Flumes and in Rivers

Olav J. M. van Duin [1,2], Suzanne J. M. H. Hulscher [1,*] and Jan S. Ribberink [1]

[1] Department of Civil Engineering, Faculty of Engineering, Marine and Fluvial System, University of Twente, 7522 NB Enschede, The Netherlands; Olav.vanDuin@deltares.nl (O.J.M.v.D.); jsribberink@home.nl (J.S.R.)

[2] Unit of Inland Water Systems, Department of Operational Water Management, Deltares, 2800 MH Delft, The Netherlands

[*] Correspondence: s.j.m.h.hulscher@utwente.nl

**Abstract:** In this paper we derive a new morphological model, with an extended version of the sediment transport model for the mean step length (the average distance travelled by sediment particles), in which this mean step length depends on the mean bed shear stress. This model makes the step length increase with increasing flow, in line with previous experimental results. To account for suspension and the large-scale turbulent structures in rivers, the step length also depends explicitly on water depth. This approach enabled modelling of the transition from dunes to the upper-stage plane bed. It was shown that by increasing the step length, the lag between shear stress and bed load transport rate increases, and the dunes eventually become smoother and lower, until finally the dunes wash out. The newly adopted model approach is tested successfully with a synthetic data set from the literature, where plane bed conditions are indeed reached in the model, similar to the results of a more advanced model. It is shown that with increasing discharge, the flow increases, which leads to higher step length and to the washing out of the dunes. Although the present model still overestimates the dune height for river cases, the potential of the model concept for river dune dynamics, including the transition to upper-stage plane bed, is shown. The model results indicate that, if a transition to upper-stage plane bed occurs in a realistic river scenario, a reduction of the water depth of approximately 0.5 m can occur.

**Keywords:** river dunes; sediment transport; morphological modelling

## 1. Introduction

Hydraulic roughness values play an important role in correctly predicting water levels in rivers [1–3], which is critical for flood management purposes. This especially holds for bifurcated rivers [4], due to feedback mechanisms between the downstream branches. River dunes increase the hydraulic roughness significantly: their shapes cause form drag. Because of their significant impact on hydraulic roughness, water level forecasts during a high river water discharge depend on accurate predictions of the evolution of river dune dimensions. One aspect of this is the correct prediction of a transition to upper-stage plane bed conditions. The general motivation of this paper is to contribute to the development of a relatively simple and physics-based dune evolution model that is able to capture proper dune dimensions and dynamic behaviour under transitional conditions. In effect, the goal is to model river dune behaviour under varying discharges between the lower-stage plane bed to the upper-stage plane bed. Such a model should work under flume and field conditions, and should predict transitions at the appropriate discharges.

In the past, many approaches have been used to model dune dimensions, varying from empirical equilibrium dune height predictors (e.g., [4–7]) to different forms of stability analyses (e.g., [8–11]). Recently, more advanced models have been developed that calculate the turbulent flow field over bedforms, in some cases in combination with morphological computations (e.g., [12–20]). These models are valuable to study detailed hydrodynamic

processes, but are computationally intensive, which makes them unsuitable for the prediction of bedform evolution and roughness over full-scale river reaches during a longer period (e.g., that of a flood wave).

To enable efficient predictions of dune dimensions over the time-scale of a flood wave, Paarlberg et al. (2009) [21] have developed a model in which the flow and sediment transport at the flow separation zone is parameterized instead of using full hydrodynamic equations. This model is able to predict the evolution of dunes from small initial disturbances up to equilibrium dimensions with limited computational time and good accuracy. In addition, this model has been coupled with an existing large-scale depth-averaged hydraulic model to form a 'dynamic roughness model' [22]. The coupled model clearly shows the expected hysteresis effects in dune roughness and water levels and different behaviour of sharp-peaked versus broad-peaked flood waves within the dune regime [22].

As Nakagawa and Tsujimoto (1980) [23] have argued, a lag distance between flow properties (and thereby bed shear stress) and sediment transport is the principal cause of bed instability and thereby regime transitions. One of the contributing factors is the step length of sediment particles, which is the distance travelled from dislodgement to rest according to Einstein (1950) [24]. This creates a phase-lag effect which is not taken into account in equilibrium transport formulas, such as that of Meyer-Peter and Müller (1948) [25]. The latter formula was used in the original model of Paarlberg et al. (2009) [21], which made it impossible to model a transition to upper-stage plane bed. The pick-up and deposition model, as proposed by Nakagawa and Tsujimoto (1980) [23], inherently allows a phase-lag effect over distance. The deposition of sediment away from the pick-up point is determined by using a probability distribution function that relies on the mean step length. This pick-up and deposition model has already been used in the dune evolution model of Shimizu et al. (2009) [14], with good results regarding prediction of generated dunes and upper-stage plane bed in flumes. However, this model is far too complicated to allow for simulations of river dune dynamics, that is, at full river scale during a longer period (e.g., a flood wave and afterwards, typically a month).

Therefore, the Paarlberg et al. (2009) [21] model was extended to enable predictions of a transition to upper-stage plane bed. Van Duin et al. (2017) [26] have shown that replacing the transport formula of Meyer-Peter and Müller (1948) [25] with the pick-up and deposition model of Nakagawa and Tsujimoto (1980) [23] leads to improved predictions of dune dimensions in the dune regime. Furthermore, this model is in principle able to simulate the washing out of dunes as well, signifying the potential of this approach for the prediction of a transition to upper-stage plane bed. Manually selecting high values of the step length leads to the washing out of dunes within this model. However, the automatic selection of a physics-based step length, and the physics behind this, is still an issue.

The model of Van Duin et al. (2017) [26] does not incorporate suspended sediment, which was clearly shown to be important for high flow conditions by Naqshband et al. (2017) [27]. Under the influence of turbulent action at the bottom of a flume or channel, sediment may be entrained in the water column, and it may therefore be necessary to allow for step length values that are higher than those typically found for bed load. Depending on the flow and turbulence structure there, sediment may be transported over a large distance in suspension. If this suspended transport occurs along dunes, the average distance travelled by sediment becomes far larger than for bed load alone (see also Yamaguchi et al. (2019) [20]). This has implications for the spatial lag between flow and sediment transport, and thereby for the river dune morphology as well, neither of which have been investigated so far.

Naqshband et al. (2017, 2014b) [27,28] used data from experiments and rivers to show that with an increasing suspension parameter, dunes first get higher (i.e., a larger amplitude to length ratio), arrive at a maximum height, then become lower, and finally disappear for large suspension numbers. These results indicate that high suspension numbers are required for the dunes to wash out, and that alongside the influence of bed load lags, suspension lags should also be included in determining dune dimensions.

One of the mechanisms contributing to this dependence of dune morphology on suspended sediment has been studied by Naqshband et al. (2014a) [29]. These authors have shown that significant portions of suspended load 'escape' the dune by not avalanching on the lee side of the dune but going over the flow separation zone and depositing further away (also shown by Kostaschuk et al. (2010) [30]). The authors find that while bed load and suspended load are comparable in magnitude, the gradient of bed load is larger than that of suspended load. This implies that the general shape of these equilibrium dunes is mostly determined by bed load, and to a lesser extent by suspended load, which deposits more evenly along the dune. However, for strongly increasing flow, turbulence becomes stronger, and suspended load can become more and more dominant. Then, the situation could arise that so much sediment is spread out evenly that dunes start decaying and may even disappear entirely (see also Engelund and Fredsøe (1974) [31], Fredsøe and Engelund (1975) [32], Smith and McLean (1977) [33], and Bridge and Best (1988) [34]). So far, these effects have not been incorporated in idealized dune evolution models (e.g., Paarlberg et al. (2009) [21]). In summary, no morphodynamic model is available that has the potential to capture the behaviour of full-scale river dunes during a flood wave from moderate to extremely high discharges and afterwards.

Therefore, in this paper we investigate the potential of the inclusion of a step length model (that combines bed load and suspended load processes, although the latter, as a first step, in a strongly schematized way) with an idealized dune model such that it enables the modelling of dune dynamics during such a flood wave. More precisely, the two research questions are: (1) To what extent can this new model replicate dune dynamics as they occur in flume conditions under variable discharge, including upper-stage plane bed and bedform hysteresis effects? (2) To what extent can this type of model in principle describe dune dynamics, including upper-stage plane bed and hysteresis effects in field conditions?

The organization of the chapter is as follows. In Section 2 the set-up of the model is discussed, while step length models from the literature and a new step length model are discussed in Sections 3 and 4, respectively. Section 5 shows the model results for flume conditions, while Section 6 shows the model results for river conditions. In Sections 7 and 8, the discussion and conclusion are presented.

## 2. Dune Model

### 2.1. General Set-Up

The basis of the present model is the dune evolution model developed by Paarlberg et al. (2009) [21]. Paarlberg et al. (2009) [21] extended the process-based morphodynamic sand wave models of van den Berg et al. (2012) [35] and Hulscher (1996) [36], with a parameterization of flow separation (see Figure 1), in order to enable simulation of finite amplitude river dune evolution. For more details on the similarities and differences of river dunes and sand waves, see Hulscher and Dohmen-janssen (2005) [37].

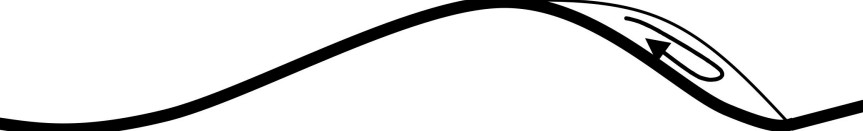

**Figure 1.** Schematization of a dune (flow left to right).

The model consists of a flow module, a sediment transport module, and a bed evolution module, which operate in a decoupled way. The model simulates a single dune which is assumed to be in an infinite train of identical dunes. Therefore periodic boundary conditions are used. Because varying discharge means the linear stability analysis as used by Paarlberg et al. (2009) [21] has to be done often during a model run (as each discharge leads a different most unstable mode, and therefore wavelength), this approach is computationally expensive. Paarlberg et al. (2009) [21] have shown that the dune length that follows from the numerical stability analysis is nearly linearly related to the water depth

(i.e., dune length = 7.3 water depth). This relationship is also in line with the empirical relations for dune height and length of Van Rijn (1984b) [7] as presented by Julien and Klaassen (1995) [38]. Therefore, for computational efficiency, this relation is also adopted in the present study.

### 2.2. Flow Model

In general the flow is forced by the difference in water level along the domain. While the water depth at the start and end of domain are the same due to the periodic boundary conditions, the water level differs because the domain is sloped. The average bed level is taken as zero and has a slope (this average bed slope is an input parameter for the model). By solving the flow equations with a certain average water depth, a discharge is found. The average water depth is adjusted until this discharge matches the discharge given as input.

The flow in the model of Paarlberg et al. (2009) [21] is described by the steady two-dimensional shallow water equations in a vertical plane (2-DV), assuming hydrostatic pressure conditions. For small Froude numbers and small vertical flow acceleration, the momentum equation in the vertical direction reduces to the hydrostatic pressure condition, and the time variations in the horizontal momentum equation can be dropped. The governing model equations that result are shown in Equations (1) and (2).

$$u\frac{\partial u}{\partial x} + w\frac{\partial u}{\partial z} = -g\frac{\partial \zeta}{\partial x} + A_v\frac{\partial^2 u}{\partial z^2} + gi \tag{1}$$

$$\frac{\partial u}{\partial x} + \frac{\partial w}{\partial z} = 0 \tag{2}$$

The velocities in the x and z directions are $u$ and $w$, respectively. The water surface elevation is denoted by $\zeta$, $i$ is the average channel slope, $g$ is the acceleration due to gravity, and $A_v$ denotes the constant vertical eddy viscosity. Note that a steady flow model is used to compute unsteady flow of a flood wave. This is a reasonable approach, because the length scale of a dune is small, so the remaining terms are large compared to the time derivatives. The computational domain is shown in Figure 2.

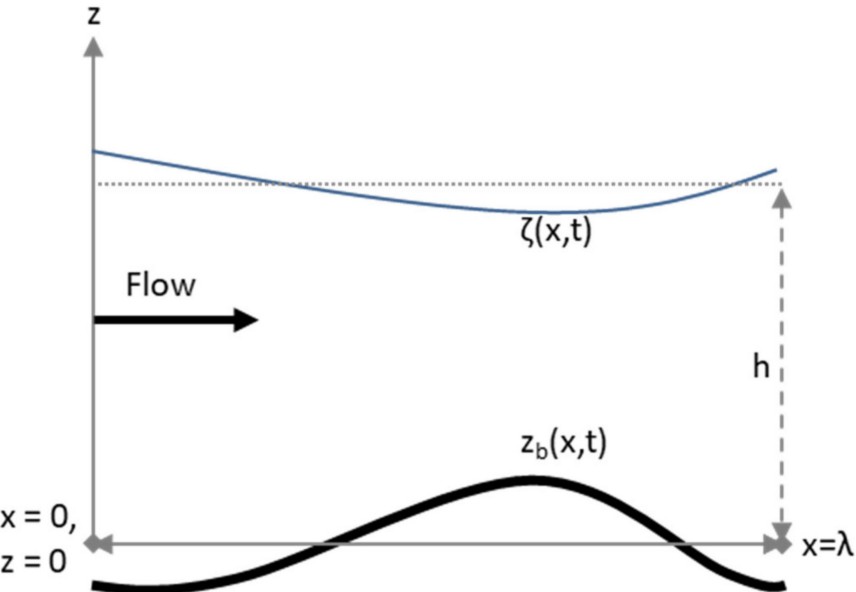

**Figure 2.** The computational domain.

In Figure 2 the symbol $\lambda$ denotes the dune length (in m), $h$ is the domain-averaged water depth (in m), and $z_b$ is the bed level relative to the x-axis (in m). The flow is forced in the domain because the x-axis is actually at a slope $i$ with regard to the real horizontal plane, creating a water level difference along the domain.

Boundary Conditions

The boundary conditions are defined at the water surface ($z = h + \zeta$) and at the bed ($z = z_b$). The boundary conditions at the water surface, Equation (3) represents no flow through the surface, and Equation (4) means no shear stress at the surface (so wind stress is assumed to be negligible). The kinematic boundary condition at the bed, Equation (5) yields that there is no flow through the bed.

$$u \left. \frac{\partial \zeta}{\partial x} \right|_{z=h+\zeta} = w \tag{3}$$

$$\left. \frac{\partial u}{\partial z} \right|_{z=h+\zeta} = 0 \tag{4}$$

$$u \left. \frac{\partial z_b}{\partial x} \right|_{z=z_b} = w \tag{5}$$

In order to close the turbulence model, a time- and depth-independent eddy viscosity is assumed. In order to represent the bed shear stress correctly for a constant eddy viscosity, a partial slip condition at the bed, Equation (6) is necessary.

$$\tau_b = A_v \left. \frac{\partial u}{\partial z} \right|_{z=z_b} = S u_b \tag{6}$$

In Equation (6), $\tau_b$ (m$^2$/s$^2$) represents the volumetric bed shear stress (i.e., without the density), $u_b$ (m/s) is the flow velocity along the bed, and the resistance parameter $S$ (m/s) controls the resistance at the bed. For more details about the model equations and numerical solution procedure, reference is made to Paarlberg et al. (2009) [21] and Van den Berg et al. (2012) [35]. For more details, the reader is referred to van Duin et al. (2016) [26].

### 2.3. Bed Load Sediment Transport Model

The pick-up and deposition model of Nakagawa and Tsujimoto (1980) [23] uses the following formulae to determine bed load transport. Pick-up of sediment (probability of a particle being picked up in s$^{-1}$) is determined by:

$$p_s(x) = F_0 \sqrt{\frac{\Delta g}{D_{50}}} \theta(x) \left[ 1 - \frac{\theta_c}{\theta(x)} \right]^3 \tag{7}$$

where $F_0 = 0.03$, $\theta$ is the Shields parameter, $\theta_c$ is the critical Shields parameter, and $\Delta = \rho_s/\rho - 1$. The grain density $\rho_s$ is set to 2650 kg/m$^3$, and the density of the water $\rho$ is set to 1000 kg/m$^3$

The local, critical volumetric bed shear stress $\tau_c(x)$, corrected for bed slope effects, is given by the following equation:

$$\tau_c(x) = \tau_{c0} \frac{1 + \eta \frac{\partial z_b}{\partial x}}{\sqrt{1 + \left( \frac{\partial z_b}{\partial x} \right)^2}} \tag{8}$$

with $\tau_{c0}$ the critical volumetric bed shear stress for flat bed, defined by Equation (9) and $\eta = \tan(\varphi)^{-1}$, in which the angle of repose $\varphi = 30°$ for sand. In this equation, $\theta_{c0}$ is the critical Shields parameter for flat bed, and $D_{50}$ is the median grain size.

$$\tau_{c0} = \theta_{c0} g \Delta D_{50} \tag{9}$$

Deposition at a location is determined by adding all the sediment that arrives at that specific location. Therefore, in order to determine the deposition at a certain location $x$, the distribution of picked-up sediment from all upstream locations is needed. The determination of deposition is done by applying the following formula:

$$p_d(x) = \int_0^\infty p_x(x - s) f(s) ds \tag{10}$$

where the distribution $f(s)$ determines the fraction of sediment that is deposited a distance s away from the pick-up point ($x - s$). The distribution function is defined as follows:

$$f(s) = \frac{1}{\Lambda} \exp\left(\frac{-s}{\Lambda}\right) \tag{11}$$

Herein $\Lambda$ is defined as the mean step length, which is further discussed in Section 2.4. By using this function, 99.3% of the sediment that has been picked up at certain location is deposited between that location and 5 times the step length in the downstream direction. The remaining 0.7% is deposited at $x = 5\Lambda$. Finally, the transport gradient is determined as follows:

$$\frac{\partial q_b}{\partial x} = D_{50}[p_s(x) - p_d(x)] \tag{12}$$

To summarize, the entire sediment transport calculation process is as follows. First, the dimensionless bed shear stress is determined from flow characteristics. Then the pick-up of sediment along the dunes follows from bed shear stress. With an exponential decay function, the deposition of sediment away from each pick-up point is determined. The difference between sediment deposition and pick-up determines the net transport gradient along the dune.

### 2.4. Step Length

To calculate how the sediment is distributed moving away from each pick-up point, the mean step length of the sediment particles has to be calculated. Step length is defined by Einstein (1950) as follows:

$$\Lambda = \alpha D_{50} \tag{13}$$

where $\alpha$ is a non-dimensional step length parameter. Francis (1973) [39], Fernandez Luque and Van Beek (1976) [40], and Sekine and Kikkawa (1984) [41] have done experiments to determine the dependency of, among other things, bed load transport, particle velocity, and step length on various parameters with moving sand along a plane bed. This data shows a range of approximately 40 to 240 times the particle diameter, for values of $u_*/w_s$ (in which $u_*$ = friction velocity and ($u_* = (\tau/\rho)^{1/2}$), and $w_s$ = settling velocity) from about 0.18 to 0.35. From this data different step length models are derived by various authors. In Section 3, two step length models based on bed load from the literature are implemented, and results are discussed. Hereafter, Section 4 presents a third method in which, in addition to the models of Section 3, suspended load processes are also explicitly accounted for. In general, the step length is assumed to be constant along the dune, so it can vary only over time due to variation in dune-averaged flow parameters.

### 2.5. Bed Evolution

The bed evolution is modelled using the Exner Equation (17), where the sediment transport gradient is calculated with Equation (12), and $\varepsilon_p = 0.4$ is the bed porosity.

$$(1 - \varepsilon_p)\frac{\partial z_b}{\partial t} = -\frac{\partial q_b}{\partial x} \tag{14}$$

After each time step for the bed evolution, the model checks the angle of the bed between every pair of neighbouring calculation points. If necessary, the model avalanches the 'excess' sand, so that the angle of the bed does not exceed the angle of repose (30°) anywhere.

## 3. Step Length Models from Literature

### 3.1. Sekine and Kikkawa (1992) Step Length Models

Sekine and Kikkawa (1992) [42] have used the data sets mentioned in Section 2 to verify a numerical model of the saltation of particles. They have found that all computed step length values are between two times larger or smaller than the observed values. Their predictions for the thickness of the saltating bed load layer closely match the data of Sekine and Kikkawa (1984) [41]; the particles remain within a few grain diameters from the bed. They further show in their calculations (1) that the mean step length varies between about 10 and about 250 times the particle diameter, (2) that it is directly proportional to

friction velocity $u_*$ ($u_* = (\tau/\rho)^{1/2}$), and (3) that it is inversely proportional with the settling velocity $w_s$. The suspension parameter $u_*/w_s$ ranges from about 0.15 to 0.28 in this set of calculations, so bed-load conditions were present ($u_*/w_s < 1$). The relation between these parameters and the non-dimensional step length $\alpha$ is as follows:

$$\alpha = \frac{\Lambda}{D_{50}} = \alpha_2 \left(\frac{u_*}{w_s}\right)^{3/2} \left(1 - \frac{u_{*c}/w_s}{u_c/w_s}\right)^{3/2} \tag{15}$$

where $\alpha_2 = 3.0 \times 10^3$, and $u_{*c}$ is the critical friction velocity ($u_{*c} = (\tau_c/\rho)^{1/2}$; note that this is *volumetric* bed shear stress, which has a unit of $m^2/s^2$).

### 3.2. Shimizu et al. (2009) Step Length Models

Shimizu et al. (2009) [14] have used a minimum ($\alpha_{min} = 50$) and maximum ($\alpha_{max} = 250$) value of non-dimensional step length $\alpha$ in a conceptually derived relation between $\alpha$ and dimensionless grain shear stress $\theta'$. For values of $\theta'$ between 0 and 0.5 (the dune regime), $\alpha$ was assumed to be constant at the minimum value ($\alpha_{min}$). For values of $\theta'$ above 0.8 (the upper-stage plane bed regime), $\alpha$ was assumed to be at the maximum value ($\alpha_{max}$). In the transitional regime ($\theta'$ from 0.5 to 0.8), $\alpha$ was linearly interpolated. Besides the Shields parameter, there is no further dependency on sediment parameters.

### 3.3. Comparison of Step Length Models

Both the conceptual model of Shimizu et al. (2009) [14] and the more physics-based model of Sekine and Kikkawa (1992) [42] are applied and tested in the dune model described in 2. For this test calculation scenario, A4 of Shimizu et al. (2009) [14] was used, which corresponds to a flood wave in a flume setting, see Figure 3. For the model runs, sediment with a $D_{50}$ of 0.28 mm, a slope $i$ of $2 \times 10^{-3}$, and a hydrograph as presented below are used.

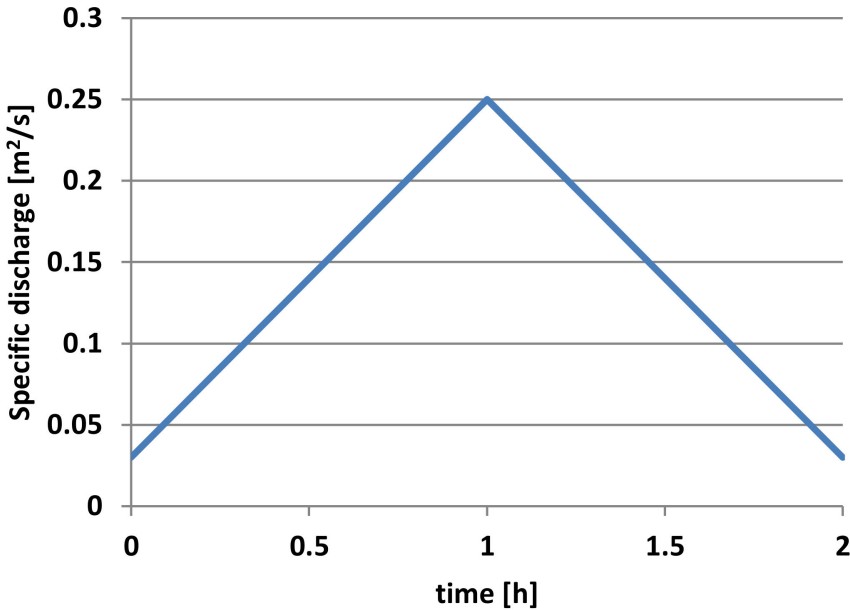

**Figure 3.** Hydrograph of scenario A4 after Shimizu et al. (2009) [14].

This artificial scenario has not been physically simulated and measured, but corresponds to flume conditions with regard to water depth and dune height. The goal of the present calculations is to investigate if transition to the upper-stage plane bed occurs in roughly the same way as the dune evolution model of Shimizu et al. (2009) [14] predicts, as well as the general qualitative behaviour of their model. This advanced dune evolution model is provided with a k-ε turbulence closure model and a separate advection-diffusion model for suspension. Scenario A4 of Shimizu et al. (2009) [14] started from a flat bed, and

showed dunes growing to a height of about 4 cm. A transition to upper-stage plane bed occurred at about 0.6 h, and a re-emergence of dunes occurred at about 1.7 h. Both step length models in the current model are tested with this scenario. The resulting development of the dune trough and crest positions (in the vertical) and water depth over time are shown in Figure 4.

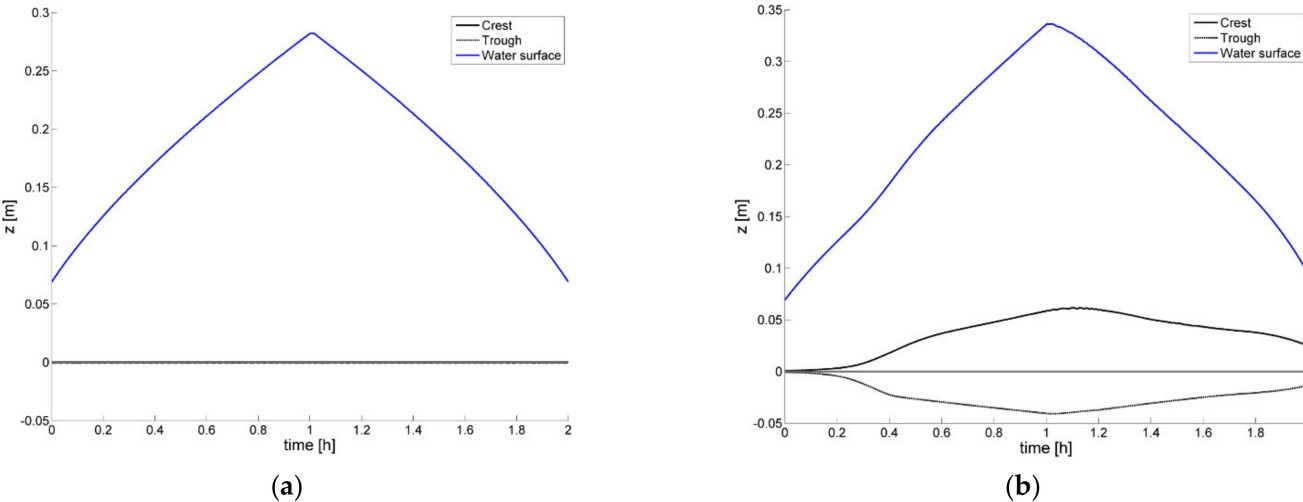

(**a**)                                      (**b**)

**Figure 4.** Dune crest and trough position (black lines) and water depth (blue line) over time with the step length model of (**a**) Sekine and Kikkawa (1992) [42] and (**b**) Shimizu et al. (2009) [14].

Using the Shimizu et al. (2009) [14] step length model leads to bedform growth, though no upper-stage plane bed occurs. Apparently, this step length model does not lead to strong enough lag in the present study; the dunes do not wash out at all but keep growing at the moment in time where the k-$\varepsilon$ dune evolution model of Shimizu et al. (2009) [14] predicted a transition to upper-stage plane bed (0.6 h). Besides growing for too long, the dunes also become too high (10 cm). With the Sekine and Kikkawa (1992) [42] step length model, no dunes are able to grow at all, which is completely different from the results of the k-$\varepsilon$ dune evolution model of Shimizu et al. (2009) [14].

### 4. New Step Length Model

Because using the Shimizu et al. (2009) [14] step length model leads to results closest to the results of the Shimizu et al. (2009) [14] dune evolution model in flumes, it was taken as the basis for further extensions to enable river dune simulations. To compensate for not modelling suspended transport explicitly, and since $u_*/w_s > 1$ for most of the scenario (it varied between 0.9 and 1.8), the parameter $\alpha$ needs to become much higher with the new step length model to implicitly take into account the effects of suspended transport. Suspended sediment load differs from bed load in the sense that turbulent vortices have the potential to move the sediment to higher parts in the water column. Due to the vertical mixing and settling process of suspended sediment, the suspended sediment load does not respond to variations in bed-shear stress (and thereby sediment pick-up) immediately, but with a spatial (and/or time) lag. To some extent, this is similar as bed-load, which also experiences a (smaller) spatial lag due to the step length of sediment grains. The turbulent mixing capacity $\varepsilon_s$ is related to eddy viscosity $\nu_t$, which for wall boundary layer flows is related to the friction velocity $u_*$ and the water depth $h$ as follows (see e.g., Van Rijn (1993) [43]):

$$\varepsilon_s \sim \nu_t \sim u_* h \tag{16}$$

The suspension mixing height above the bed is controlled by the turbulent mixing capacity $\varepsilon_s$, and the settling by gravity (settling velocity $w_s$) and scales with $\varepsilon_s/w_s = u_* h/w_s$ (Rouse, 1937) [44]. Galappatti and Vreugdenhil (1985) [45] derived a depth-averaged model for suspended sediment based on the usual advection-diffusion equation. They show that the vertical processes of turbulent diffusion and settling of suspended sediment can be

translated into a relaxation or adjustment process of suspended sediment in flow direction, involving an adjustment length (or spatial lag) of suspended sediment. A generalized expression is as follows:

$$\Lambda_{s;l} = \text{ function } \left( \frac{u_*}{w_s} \right) \cdot h \tag{17}$$

The equation shows that the spatial lag directly scales with the water depth $h$. The process of turbulent diffusion is represented by the friction velocity $u_*$, the settling process by the settling velocity $w_s$.

In addition, Claudin et al. (2011) [46] derived a similar relation for the relaxation length of suspended sediment.

There is a difference in how the spatial lag of bed-load and suspended load relate to the water motion: where the step length of bed load depends on (bed) friction velocity, the relaxation length of suspended sediment scales with friction velocity and the water depth (see Equation (17)). Therefore, as a first step, it is assumed in the present study that the step length of particles $\alpha$ should also scale with the water depth $h$ in order to simulate the influence of suspended sediment. A new model for non-dimensional step length $\alpha$ is proposed by the following equation, which depends on both non-dimensional grain shear stress and water depth.

$$\alpha \left( \theta', h \right) = \alpha_g \left( \theta' \right) \frac{h}{h_{ref}} \tag{18}$$

Here $h_{ref}$ is a reference water depth equal to the water depth at the start of the transitional regime of the case used to tune the step length model, scenario A4 of Shimizu et al. (2009) [14], of which the results can been seen in Figure 5. The value of the non-dimensional grain shear stress-dependent step length $\alpha_g$ follows from a modified version of the Shimizu et al. (2009) [14] step length model, as can be seen in Figure 5. To reiterate, with this modification the step length no longer depends on only bed shear stress or friction velocity as with bed load, but also on the water depth as with suspended transport. It represents processes inherent in the turbulent mixing of suspended material—namely, that larger water depth leads to larger turbulent vortices, which in turn leads to sediment higher in the water column, and due to the size of the vortices, a larger settling distance.

Furthermore, in the new step length model the dimensionless step length keeps increasing for $\theta'$ values above 0.8 (with the same slope as between $\theta' = 0.5$ and $\theta' = 0.8$) because formulae for the spatial lag or adjustment length of suspended sediment are generally not capped at a certain value (see e.g., [45,46]). Different values of $\alpha_{min}$ at $\theta' = 0.5$ and $\alpha_{max}$ at $\theta' = 0.8$ have been tested, and using $\alpha_{min} = 50$ and $\alpha_{max} = 350$ works best within the new dune evolution model compared to the dune evolution results of Shimizu et al. (2009) [14]. The water depth at the start of the transitional regime was 0.1166 m, which will be used as the reference water depth $h_{ref}$. In Figure 5 the currently used model for $\alpha_g$ is compared with the Shimizu et al. (2009) [14] step length model for $\alpha_S$. It should be noted that this figure is without the additional influence of the water depth as defined in Equation (18). Combining that equation with the figure above and all other previous considerations leads to the following new equation for $\alpha$:

$$\alpha \left( \theta', h \right) = \begin{cases} \alpha_{min} \frac{h}{h_{ref}}, & |\theta' \leq 0.5 \\ \left[ \alpha_{min} + (\theta' - 0.5) \frac{\alpha_{max} - \alpha_{min}}{0.8 - 0.5} \right] \frac{h}{h_{ref}}, & |\theta' > 0.5 \end{cases} \tag{19}$$

where $\alpha_{min} = 50$, $\alpha_{max} = 350$, and $h_{ref} = 0.1166$ m.

By multiplying the result with the median grain diameter, the dimensional step length is found (as defined by Equation (13)). By using this method, the value of $\alpha$ can become higher than the maximum observed value of $\alpha$ (250) found in the bed load experiments of Nakagawa and Tsujimoto (1980) [23].

With this modified method for step length calculation, the specific influence of suspended sediment on the spatial lag of sediment transport (i.e., through the water depth) is introduced in a strongly schematized way. This leads to the possibility of also applying the

model in the full-scale conditions of rivers (with larger water depths), in which suspended sediment is known to play a large role.

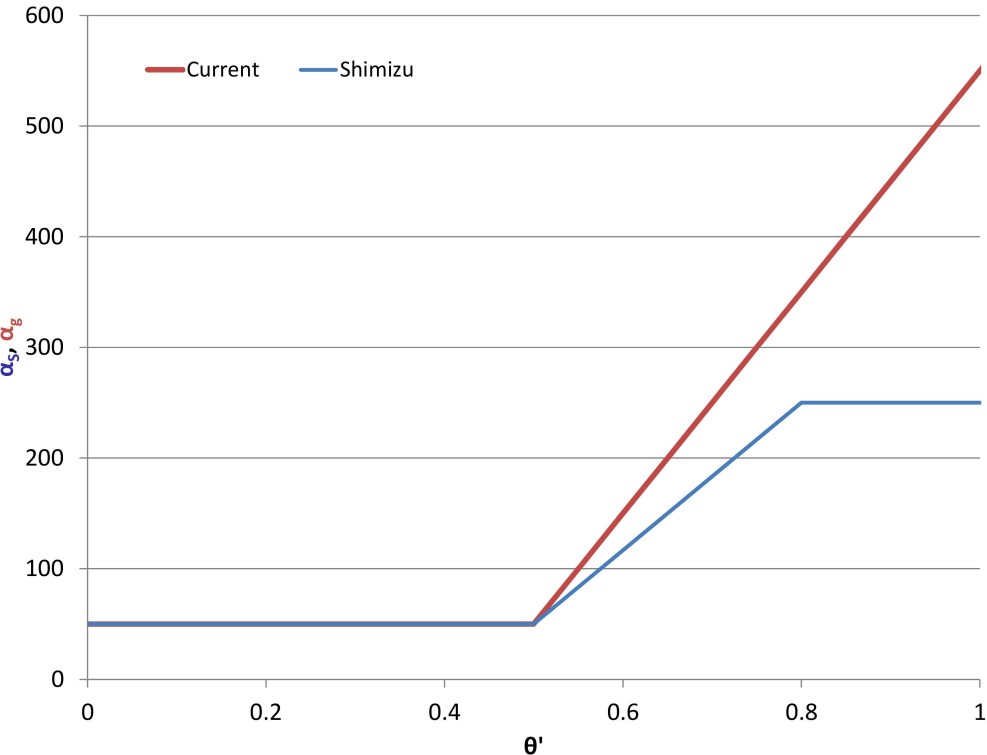

**Figure 5.** The non-dimensional step length of the non-dimensional step length model of Shimizu et al. (2009) [14], using $\alpha_S$ for bed load, and the currently used step length model Equation (19), using $\alpha_g$, which represents the grain-shear-stress-dependent part of non-dimensional step length in the new model. In this figure the depth is assumed to be equal to $h_{ref}$ in order to more clearly show the relation with the Shimizu step length model.

## 5. Results with Flume Conditions

We start with testing whether the model is able to model a transition to upper-stage plane bed for flume conditions, that is, the computational scenario A4 as presented by Shimizu et al. (2009) [14] (see also Section 3.3). With this scenario, Shimizu et al. (2009) [14] show that their model is able to predict transitions to the upper-stage plane bed. The model also clearly shows hysteresis effects; the relation between discharge and water depths is as significantly different for the rising limb of the hydrograph as it is for the falling limb. The parameters of the scenario are typically equivalent to a flume scenario. With the model runs presented here, the suspension parameter $u_*/w_s$ varied between 0.9 and 1.8 during the scenario, so with this sediment and this flow regime the suspension regime is present for most of the time. Using scenario A4 from Shimizu et al. (2009) [14], the following development of the dune field over time is found with the dune evolution model as developed in the present study.

Here in Figure 6 the washing out and regrowth of the dunes can be clearly observed. From the same results, the development of the dune trough and crest positions (in the vertical) and water depth over time are shown in Figure 7. As can be seen in the beginning of the run, dunes start developing along with increasing discharge. At a certain high discharge, the shear stress and the step length become so high that the dunes are washed out. Due to the decrease in form drag and thereby *total shear stress*, the water level stabilizes temporarily, despite still rising discharge. The bed remains washed out until the discharge and step length become so low that dunes start developing again.

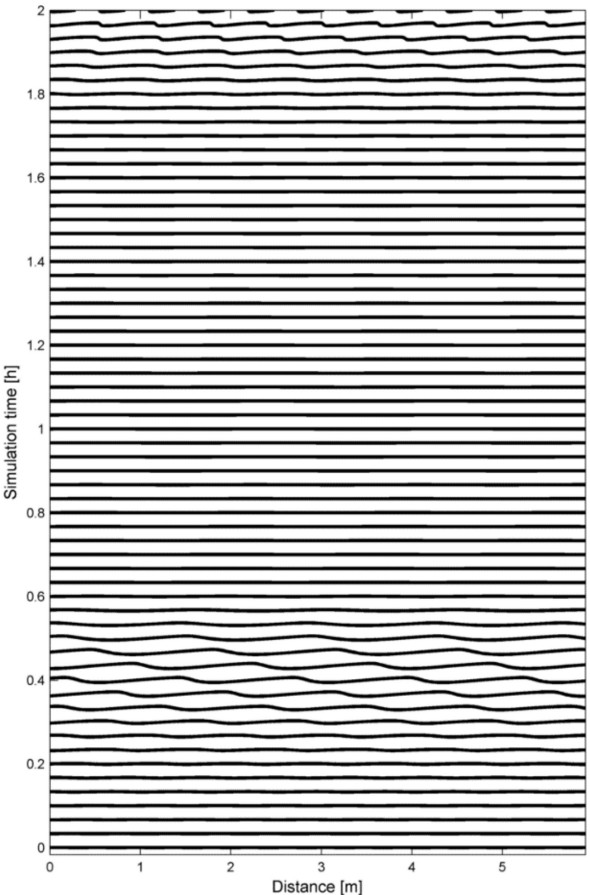

**Figure 6.** Dune field over time, using scenario A4 from Shimizu et al. (2009).

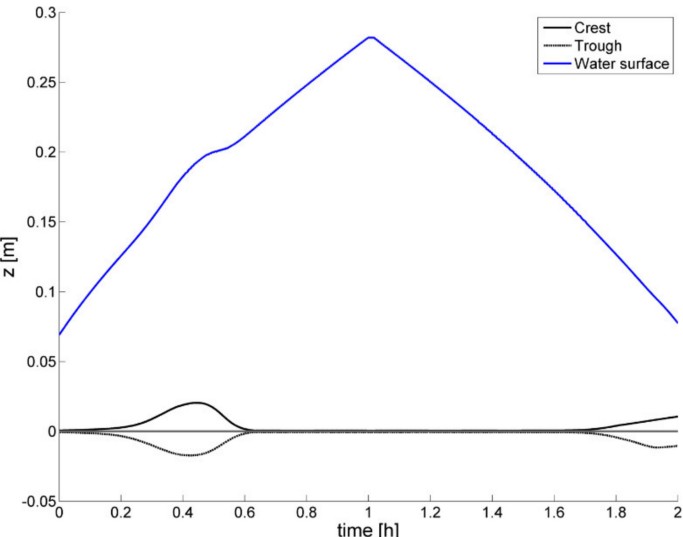

**Figure 7.** Dune crest and trough position (black lines) and water depth (blue line) over time, using scenario A4 from Shimizu et al. (2009).

Shimizu et al. (2009) [14] found roughly the same moments of washing out (0.6 h) and re-emergence of dunes (1.7 h). Furthermore, the dune evolution model of Shimizu et al. (2009) [14] predicted a dune with a maximum height of about 4 cm, which corresponds rather well with the present model result (3.75 cm). The relation between the step length parameter $\alpha$ and the specific discharge for the rising and falling stages of the hydrograph can be seen in Figure 8.

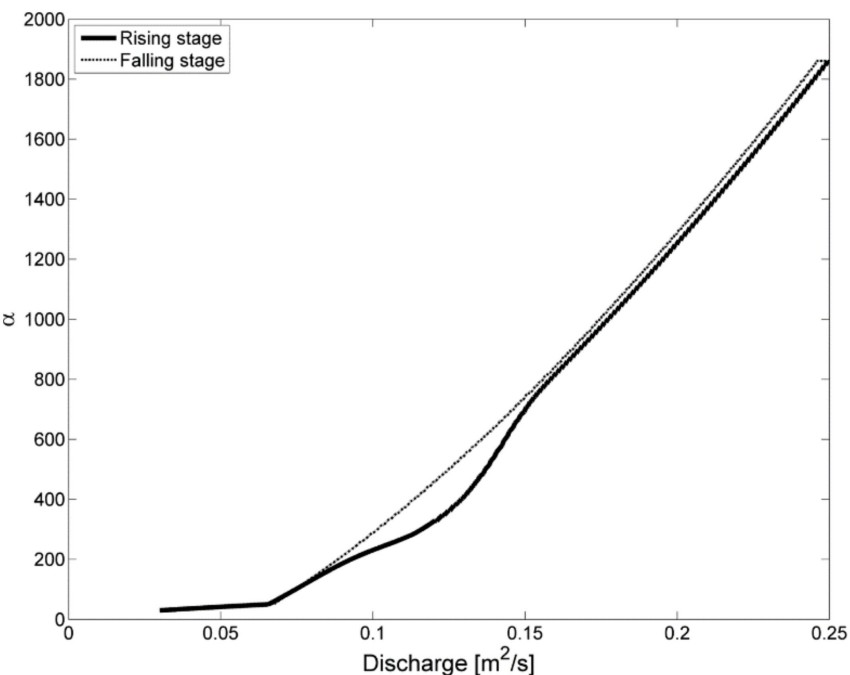

**Figure 8.** Specific discharge versus non-dimensional step length $\alpha$ separated for the rising and falling stages of the hydrograph.

Here a hysteresis effect is already observed, caused by the transition to upper-stage plane bed and then later returning to the dune regime. With the same discharge, dunes can be present in the rising stage and not in the falling stage. Because of the presence of dunes, a part of the flow power is lost due to form drag. This means the shear stress acting on the sand grains (effective bed shear stress) is relatively lower, and a lower step length is selected than when there are no dunes present. The effect of hysteresis on water depth can be seen in Figure 9.

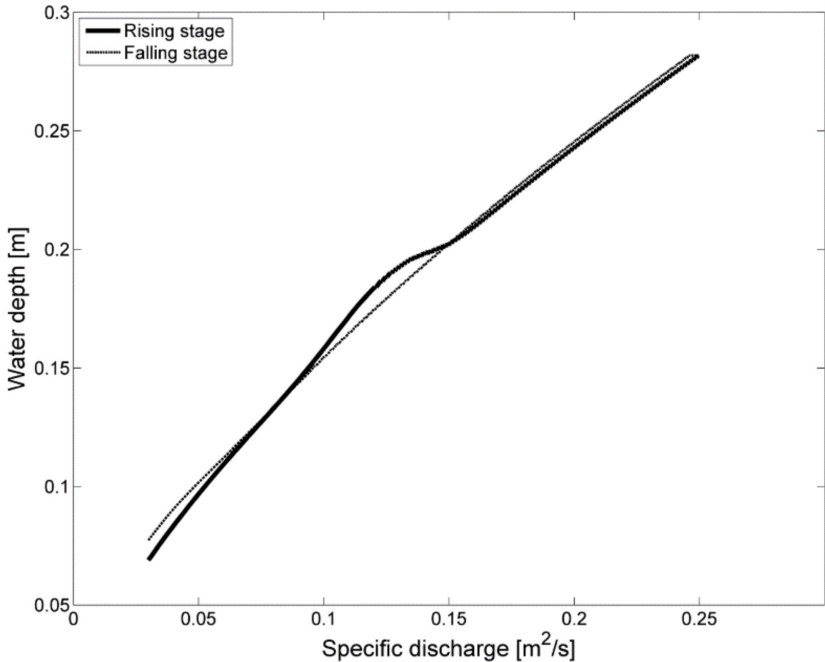

**Figure 9.** Specific discharge versus water depth separated for the rising and falling stages of the hydrograph, using scenario A4 from Shimizu et al. (2009).

In the rising part of the hydrograph, dunes are able to develop firstly at medium discharges, while in the falling part, the dunes start developing at the end of the flood wave at lower discharges. This has a clear effect on the resulting water depths at the same discharge. For example, at a specific discharge of 0.12 m²/s, the water depth in the rising part is clearly higher than in the falling part. In the rising limb, the dunes have had a longer time to grow than in the falling limb, and are therefore higher. Because the dunes are higher, the water depth is higher despite the discharge being the same. Shimizu et al. (2009) [14] have clearly shown this hysteresis effect as well, though for their model it's more pronounced. They have also reported in the order of 25% lower water depths than we found.

## 6. Results with River Conditions

Although the new dune evolution model is still in a developing stage, from a practical point of view it is interesting to test how the model behaves under river conditions. For this purpose the model is run with similar conditions to the Dutch river Waal during the flood of 1998. At the peak of this flood wave, the total discharge was 6250 m³/s in the river Waal [47]. For simplicity, the dune evolution model is applied to only the main channel; interactions with the floodplains are not taken into account here. It is assumed that 60% of the peak discharge went through the main channel. The main channel is assumed to be 300 m wide, which corresponds well to the width of the main channel along the river Waal in the SOBEK model made by Deltares and used by Paarlberg (2012) [48]. This would make the specific discharge 12.5 m²/s, which is rounded to 13 m²/s for the current study. To simulate these conditions, a $D_{50}$ of 1.2 mm (in the range presented by Giri et al., 2008 [49]), a slope $i$ of $1 \times 10^{-4}$ [50], and a hydrograph as presented in Figure 10 are used as input for the model. The settling velocity corresponding to the used $D_{50}$ is $w_s$ = 0.13 m/s. This hydrograph leads to water depths between 8.5 and 11.1 m, which correspond well to the typical water depths in the Waal for the flood of 1998 as reported by Julien et al. (2002) [47].

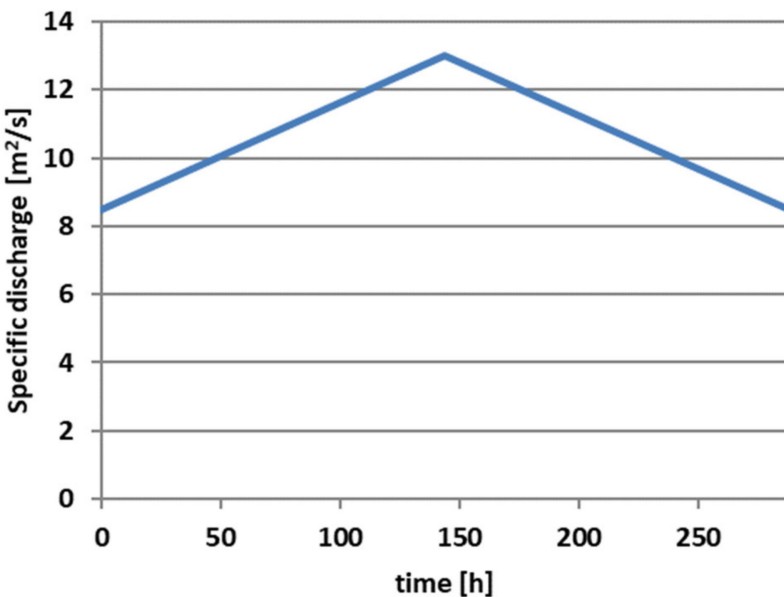

**Figure 10.** Hydrograph of the river scenario, which is based on the 1998 flood in the river Waal.

Compared to the flood wave of 1998, as shown by Julien et al. (2002) [47], we kept the duration of the rising limb the same, while we shortened the duration of the falling limb somewhat in order to create a symmetric flood wave. The entire duration of the flood wave presented here is exactly 12 days or 288 h. The calculation time is very reasonable; with a time step of 2 min, the model goes through 12 days modelled time in under 55 min real time. To let the model adjust to the hydrodynamic conditions, the flood wave is modelled

twice. The bed configuration of the first flood wave is used as input for the second flood wave. The results of the second run are reported in Section 6.1.

To test if a transition to the upper-stage plane bed can be modelled for river conditions, the model is also run with a 5% higher discharge than for the scenario above. This leads to water depths between 9 and 11.4 m, which is about 5% higher than the water depths in Waal for the flood of 1998 as reported by Julien et al. (2002) [47]. The duration of the flood wave is the same as in the other scenario, and the calculation time is just under 35 min. This is less than before, because in this scenario dunes are washed out during a part of the run, which lets the model solve the flow equations more efficiently. Again, to let the model adjust to the hydrodynamic conditions, the flood wave is modelled twice. The results of the second run are reported in Section 6.2.

### 6.1. Results for the River Scenario

The development of the dune field, starting from an already developed dune field, can be observed in Figure 11.

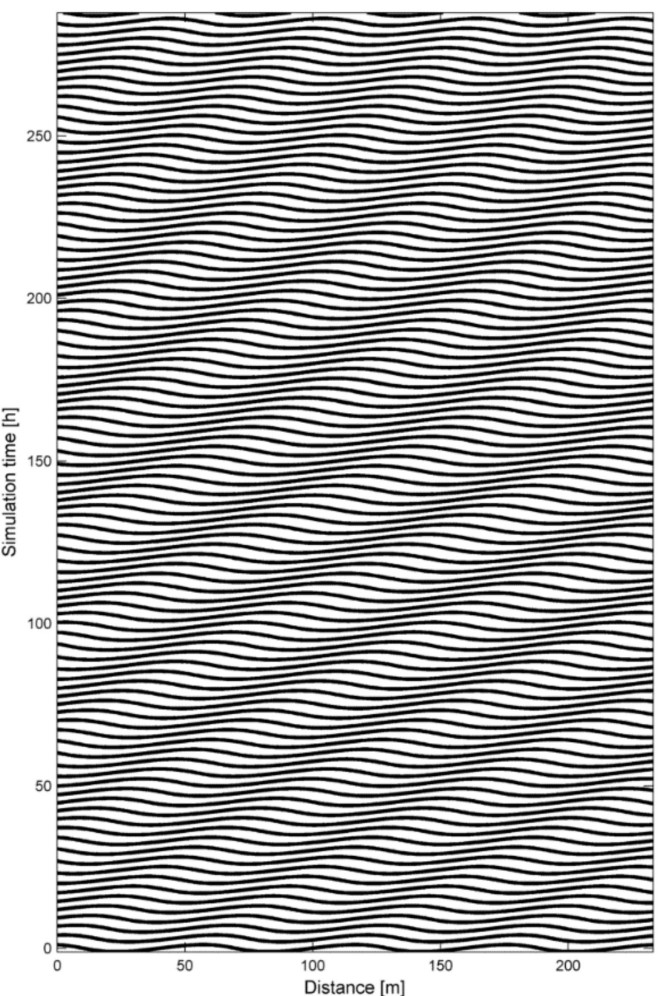

**Figure 11.** Dune field over time, using the river scenario.

The dunes grow and shrink during the flood wave, though in general the dune height variation is low. From the same results, the development of the dune trough and crest positions (in the vertical) and water depth over time are shown in Figure 12. As can be seen in the beginning of the run, dunes start developing slowly along with increasing discharge. This continues past the moment of peak discharge until about 155 h, after which the dune height decreases. Due to the increase in discharge, the water level still increases and decreases, with seemingly little influence of the dune height.

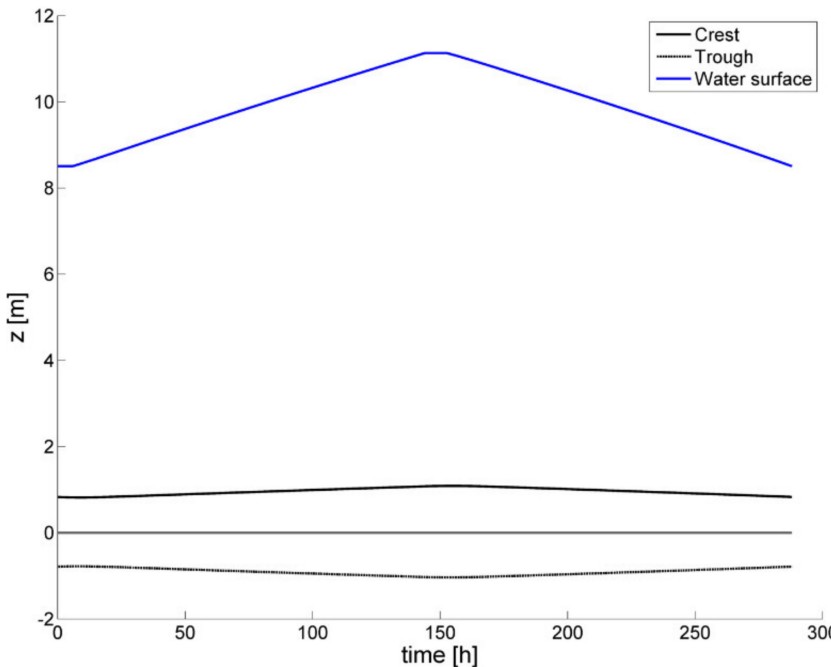

**Figure 12.** Dune crest and trough position (black lines) and water depth (blue line) over time, using the river scenario.

During the flood wave of 1998, dune heights in the Waal varied between 0.2 and 0.6 m on the centreline [47], while the model predicts values between 1.6 and 2.1 m. This is still a considerable difference, which indicates that further model improvements are necessary. Increasing the step length likely has a dampening effect on the dune height, but it also causes the dunes to wash out sooner. Therefore, care should be taken in adjusting the model settings. In Figure 13 the response of the water depth to the varying discharge can be seen.

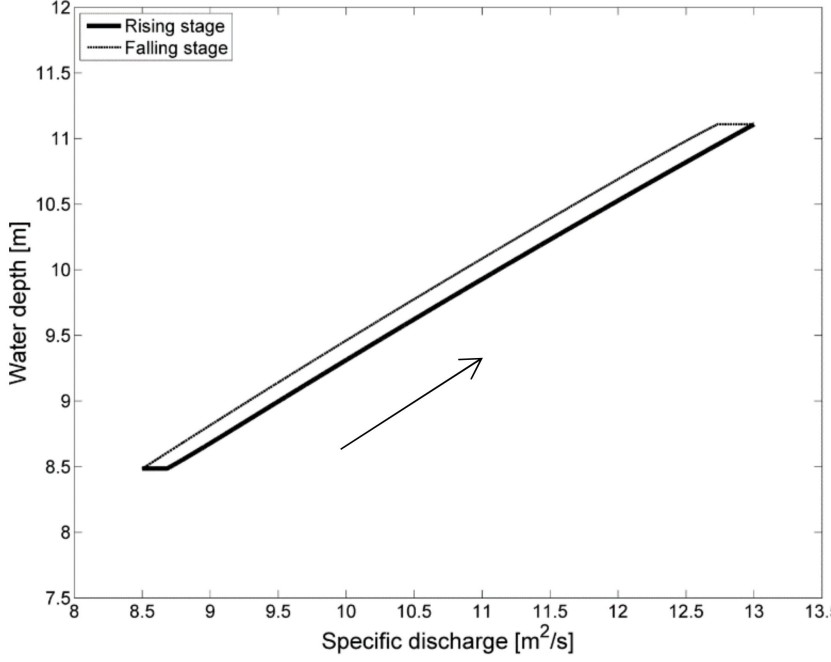

**Figure 13.** Discharge versus dune height for the rising and falling stages of the hydrograph, using the river scenario. The arrow signifies the direction of development over time.

The hysteresis loop observed here is in the same direction as for the flood wave of 1998 as reported by Julien et al. (2002) [47], with dune heights in the falling limb being higher than in the rising limb (Figure 14). Here the dune height also keeps growing for a period of time after the discharge is already decreasing. The relation between the non-dimensional shear stress and the dune height can be seen in the following figure. The dunes do not wash out; however, they do decrease in height about 5 h after the maximum discharge has been reached.

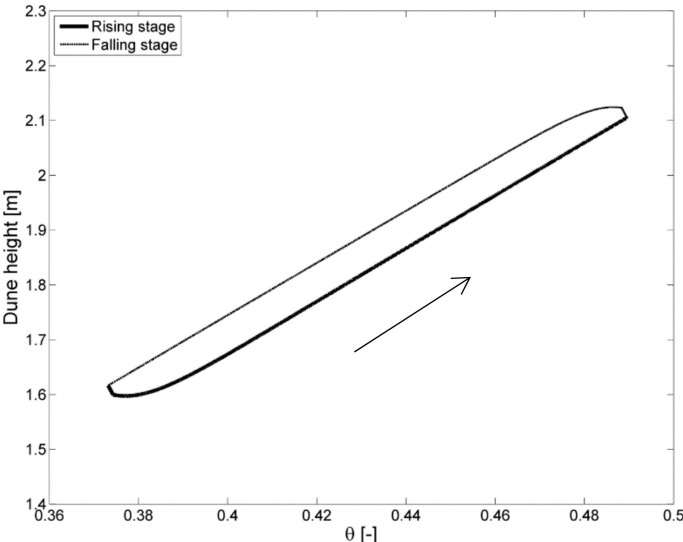

**Figure 14.** Non-dimensional shear stress $\theta$ versus dune height for the rising and falling stages of the hydrograph, using the river scenario. The arrow signifies the direction of development over time.

The relation between $\alpha$ and the specific discharge for the rising and falling stages of the hydrograph can be seen in Figure 15. Here again a hysteresis effect is observed, caused by the lag between dune development and discharge. It should be noted that now, much higher step lengths are reached than for the flume scenario, up until $\alpha = 4750$, which is a dimensional step length of about 6 m. This is about 2.6 times higher than that of the flume scenario.

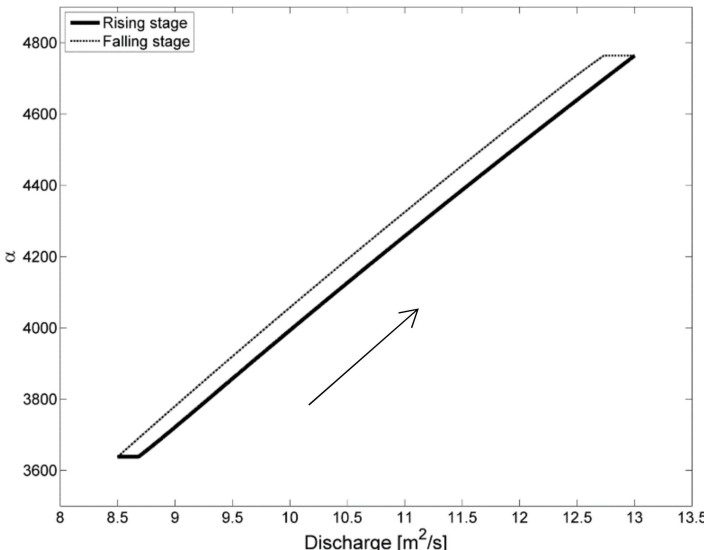

**Figure 15.** Specific discharge versus non-dimensional step length $\alpha$ separated for the rising and falling stages of the hydrograph, using the river scenario. The arrow signifies the direction of development over time.

The effect of hysteresis on water depth can be seen in Figure 16.

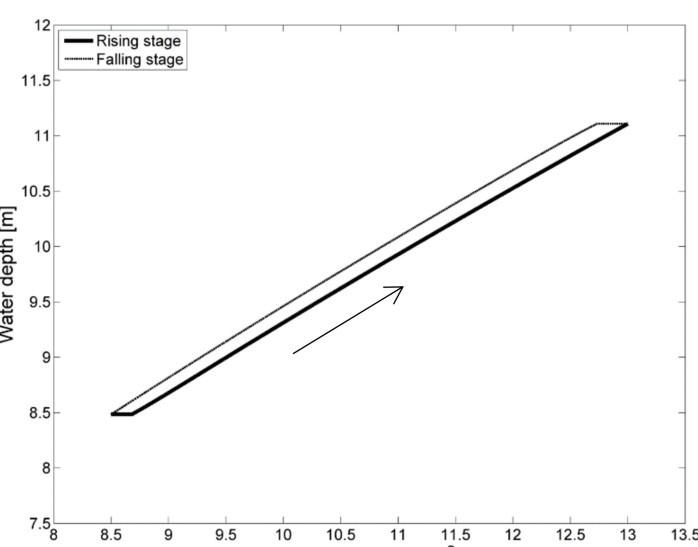

**Figure 16.** Specific discharge versus water depth separated for the rising and falling stages of the hydrograph, using the river scenario. The arrow signifies the direction of development over time.

In the rising part of the hydrograph, dunes increase in height, while in the falling part the dune height first remains constant before decreasing later on. Because of the delay in dune height development compared to discharge, the water depth in the falling limb of the hydrograph is larger than in the rising limb, which agrees again in a qualitative sense with what is reported by Julien et al. (2002) [47] for the flood wave of 1998.

### 6.2. Results for the More Extreme River Scenario

With the artificial more extreme scenario, which includes an increase of the discharge of 5%, a transition to upper-stage plane bed is predicted. It should be noted that this transition was not observed in the river Waal in 1998 [47], nor in the Dutch Rhine Branches in 1995 when the discharge at Lobith was 25% higher than it was in 1998 (e.g., [51]). The computed development of the dune field, starting from an already developed dune field, can be observed in Figure 17.

Here the growth and washing out of the dunes can be observed on the left, and the washing out itself can be observed more clearly on the right. From the same results, the development of the dune trough and crest positions (in the vertical) and water depth over time are shown below (Figure 18). As can be seen in the beginning of the run, dunes start developing along with increasing discharge. At around 137 h from the start of the model run, the dunes start washing out, and they're completely gone at about 141 h, 5 h before the maximum discharge occurs. Due to the decrease in form drag and thereby *total shear stress*, the water level goes down, despite still-rising discharge. The bed remains washed out until the discharge is low enough for dunes to start developing again.

For the flood of 1998, dune heights in the Waal varied between 0.2 and 0.6 m on the centreline, while the model results indicate values between 0.2 and 2.2 m. In Figure 19 the response of dune height to changing discharge can be seen.

The hysteresis loop observed here is in the opposite direction of the measured flood wave of 1998 as reported by Julien et al. (2002) [47]. Due to the washing out of dunes during the flood wave, dunes are now generally lower in the falling limb than in the rising limb.

The relation between $\alpha$ and the specific discharge for the rising and falling stages of the hydrograph can be seen in Figure 20.

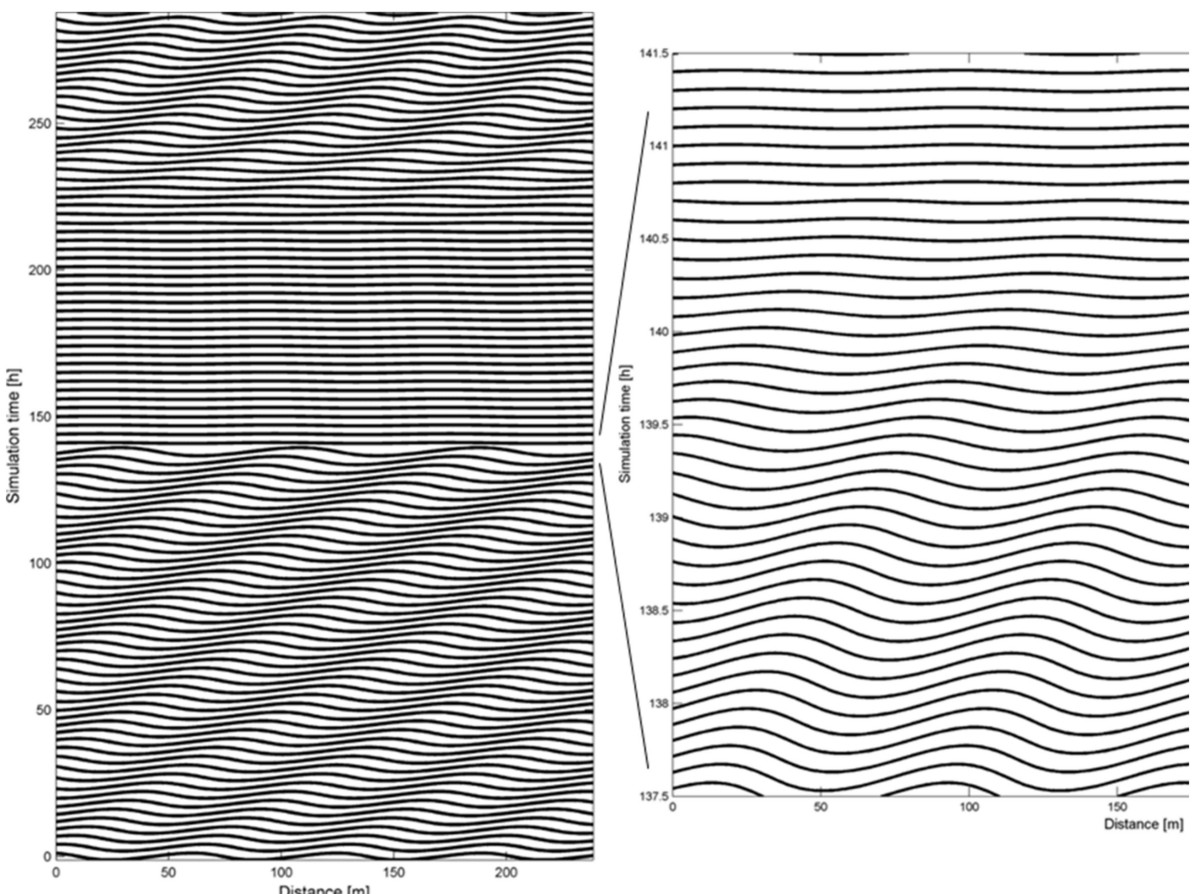

**Figure 17.** Dune field over time, using the more extreme river scenario. The right part zooms in on the dune field in the period where the transition occurs.

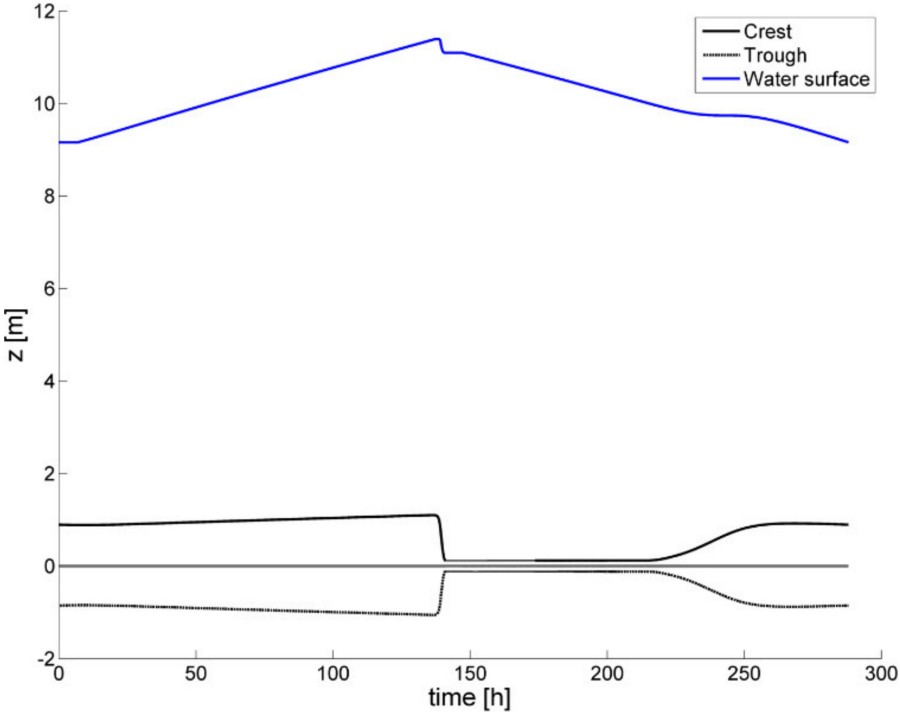

**Figure 18.** Dune crest and trough position (black lines) and water depth (blue line) over time, using the more extreme river scenario.

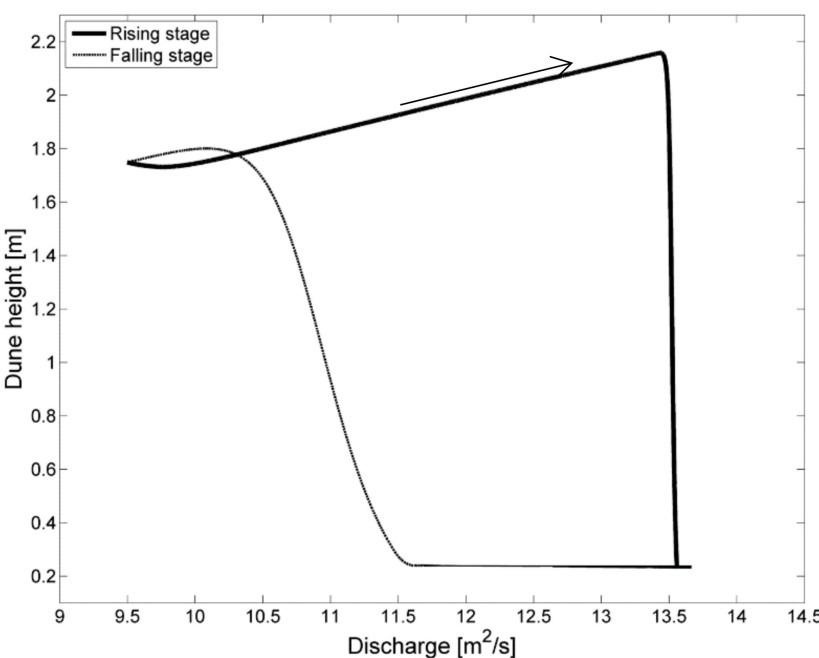

**Figure 19.** Discharge versus dune height for the rising and falling stages of the hydrograph, using the more extreme river scenario. The arrow signifies the direction of development over time.

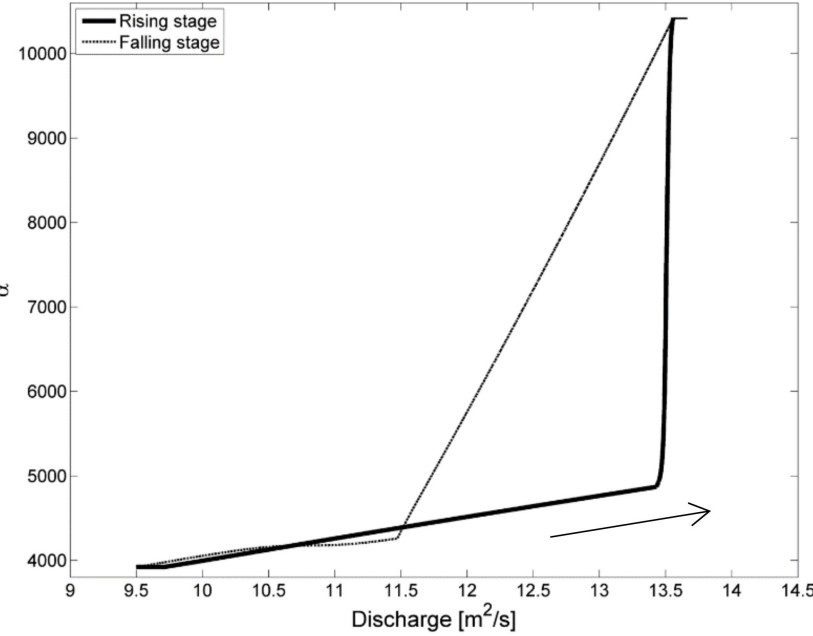

**Figure 20.** Specific discharge versus non-dimensional step length $\alpha$ separated for the rising and falling stages of the hydrograph, using the more extreme river scenario. The arrow signifies the direction of development over time.

Here again a hysteresis effect is observed, caused by the transition to upper-stage plane bed and back to the dune regime. It should also be noted that now, much higher step lengths are reached, up until $\alpha = 10,800$, which is a dimensional step length of about 13 m. This is about 10 times the value of the flume scenario. For the river scenario of Section 6.1, the maximum step length was only 2.6 times higher than the flume scenario. This difference between the river scenarios mostly relates to the washing out of dunes, which only occurs for the more extreme river scenario. When the dune is washed out, water depth drops somewhat, but due to the lack of bedforms, the flow can exert more

force on the particles, which increases step length because that is partly based on grain shear stress. The effect of hysteresis on water depth can be seen in Figure 21.

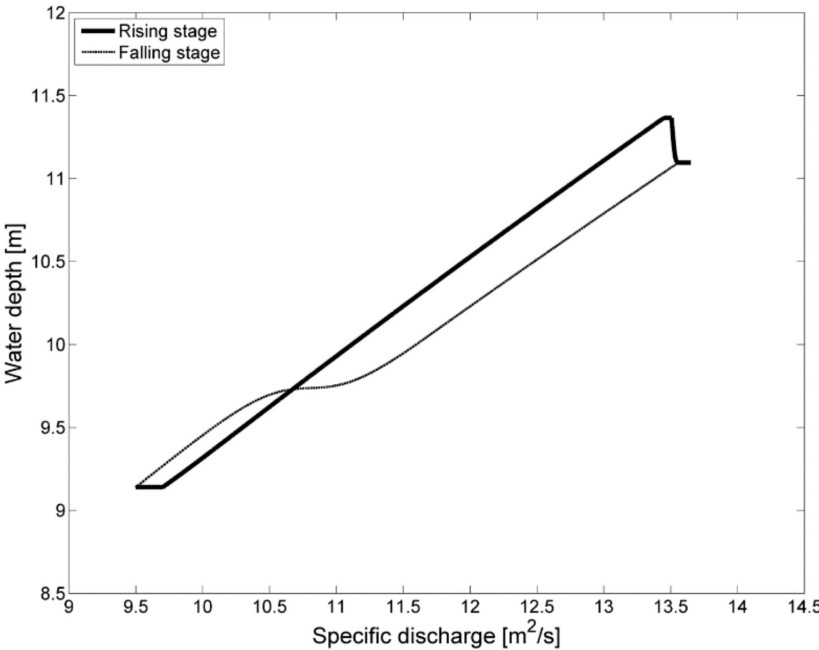

**Figure 21.** Specific discharge versus water depth separated for the rising and falling stages of the hydrograph, using the more extreme river scenario.

In the rising part of the hydrograph dunes are able to develop firstly at lower discharges, while in the falling part the dunes start developing at higher discharges. This has a clear effect on the resulting water depths at the same discharge. The hysteresis loop is reversed compared to the results in Section 6.1. Of great importance for the river situation is the significant drop in water levels when dunes are washed out. Within 4 h the water levels become about 0.25 m lower, while the discharge is still growing; if it assumed dunes keep growing instead, and we extrapolate from the point where the water depth is still rising, the difference in water levels is approximately 0.5 m. Though this effect will probably be mitigated in real life by the presence of floodplains, it is still a significant effect. This example shows that knowing when the transition to upper-stage plane bed occurs is important, because it may have a large impact on the water levels and thereby on the dike heights needed to prevent flooding.

## 7. Discussion

In the present study, under flume conditions, differences have been observed in outcome between the new model and the k-ε model of Shimizu et al. (2009) [14], that is, step lengths have to become significantly larger to reach the transition to an upper-stage plane bed, and there are differences of 25% in water depths. Of course, the models differ in significant ways. Firstly, Shimizu et al. (2009) [14] have used a non-hydrostatic flow model with a non-linear k-ε model for turbulence closure, whereas in the idealized dune evolution model, a hydrostatic flow model with a constant eddy viscosity has been used. Secondly, their transport module also used a separate suspended sediment model, while in this research suspended sediment has been only modelled implicitly. Pinpointing the exact reason of the differences is hard because the A4 flume case under review is a 'synthetic' case; there are no actually measured values as a reference to determine which model is closer to the truth. The Shimizu et al. (2009) [14] dune model is more physically complex, which suggest that its results should be better. However, it is still promising that the new relatively simple model presented in this research is able to represent similar bedform behaviour during a flood wave as the more complex model in a qualitative sense.

During the research for this paper, it was found that the timing of a transition to upper-stage plane bed is sensitive to the value of $\alpha$. As an example, the results of the first 45 min for scenario A4 of Shimizu et al. (2009) [14] with $\alpha_{max} = 300$ as well as $\alpha_{max} = 350$ are shown in Figure 22. Here it can be seen that the bed is washed out at about 37 min with $\alpha_{max} = 300$ instead of 35 min as with $\alpha_{max} = 350$, and that the maximum dune height is 3.75 cm and 2.8 cm, respectively. This shows that the results are sensitive to the settings of the step length model, though there is no extreme effect on general model behaviour.

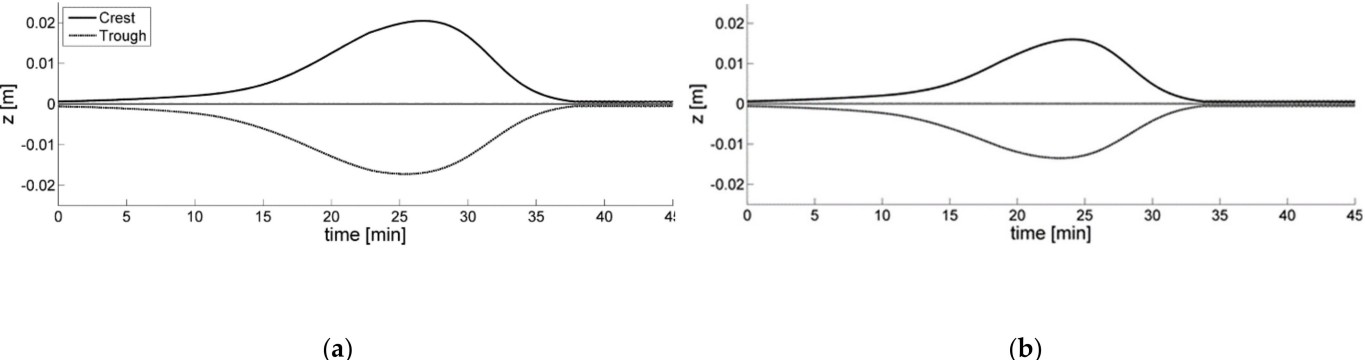

(**a**)                                                    (**b**)

**Figure 22.** Dune crest and trough position for the first 45 min of scenario A4 of Shimizu et al. (2009) [14]) with (**a**) $\alpha_{max} = 300$ and (**b**) $\alpha_{max} = 350$.

To ensure that the model still represents the dune regime well, the model was run again for the experimental conditions of Venditti et al. (2005a, 2005b) [52,53] tested in the Section Results of van Duin et al. (2017) [26]. Compared to the previous results, the dune height is now generally slightly lower, though still represented well. Dune length is very similar to before, represented well except for the artificial defects of bedforms D and E. Migration rate and thereby transport rate are much higher than before, and are now generally overestimated instead of underestimated. This is in line with the results presented in Section 3.3 of van Duin et al. (2016) [26], where the migration rate and thereby transport rate increased strongly with step length. Overall the results are slightly worse than with the model version optimized for the dune regime, though dune dimensions are still represented well by the new model.

The question arises whether the step length should be varied only due to the changing flow regime as is done now, or along the dune as well (because of local variation in shear stress). Explorative experiments of Van Duin et al. (2012) [26] suggest that the latter is not necessary, but this warrants further research. This also applies for more realistic non-linear relations of the step length model, as the current one (Equation (19)) is still relatively simple.

Although the model results show hysteresis effects with regard to discharge and water depth caused by the development of dunes, it should be realized that the present model does not describe the hysteresis effects due to the phase lead of discharge with respect to water depth, which may be relevant for field situations. This phase lead arises from the phenomenon that the pressure gradient of the front of the flood wave is higher than in the tail of the flood wave, as described by Jones (1915) [54] and later studied by various authors (e.g., [55–57]).

The spatial phase lag of the sediment transport relative to the local bed shear stress is the crucial element of the model for representing the growth and washing out of bedforms. This sediment transport phase lag is not an explicit part of our model, but it is brought into the model through the step-length of grains. This stepping of grains leads to the situation that the local sediment transport along the dune is coupled to the bed shear stress (and grain pick-up) some distance upstream. The original Einstein step length model in principle is a bed load formulation, as it explicitly models grains to make a step at a distance of some grains (i.e., rolling or sliding a small distance). To incorporate the suspension effect, we are stretching the Einstein step length model towards much larger steps. For

suspended sediment, this phase-lag is caused by the relatively slow process of upward turbulent diffusion and downward settling of suspended sediment particles. The length scale of this suspension adjustment process is a function of the suspension numbers $u_*/w_s$ and increases with the water depth $h$ (see Equation (17)). Now, by incorporation of the depth in the step length formulation, we allow for much larger steps. Such larger steps can be thought of as overflowing the flow separation zone in a dune, so that the sediment accumulates at the stoss side of the next dune. If this step becomes very large, suspended sediment will be able to pass even more than one dune crest.

It could be interesting to use a separate suspended transport model similar to the bed load models used in this study, with the lower step lengths used for bed load and the higher step lengths used for suspended load. With such a dual method, it is important to know what the appropriate step lengths are for each of the transport modes. This could be done by either implementing more physics-based predictors of the step length for bed load and suspended load, or by calibrating the step length models using dune data sets from the field, such as that of Sieben (2004) [50], as a benchmark. Another interesting exercise would be to test the Froude number effect in this model with laboratory and field experiments (see e.g., Naqshband et al. (2014) [28]).

In general, a well-calibrated version of the dune evolution model could then be applied within a model of a river system (replacing a part of the hydraulic roughness model) to better determine if and under which conditions a transition to upper-stage plane bed may occur. As far as we know for the Dutch rivers, washing out of dunes with a high discharge has never occurred, and it is unknown what will happen with the extreme (design) discharges that may occur in the future. In current practice, roughness values are calibrated by matching observed and modelled water levels [58,59]. This means that there is significant uncertainty in roughness for extreme discharges that have not been observed yet. Using a dynamic roughness model which includes the dune dynamics may lead to more physically correct results. The river Waal case in this study for the 1998 flood has shown that a relatively small increase in river discharge can lead to the occurrence of upper-stage plane beds together with a considerable drop in water depth in a relatively short time. This shows that the occurrence of upper-stage plane bed can be an unexpected and rapid occurrence.

## 8. Conclusions

In this study the dune evolution model of Paarlberg et al. (2009) [21] was extended with an alternative sediment transport model using sediment pick-up, deposition, and step length models instead of the usual equilibrium transport formula. The model was devised in such a way that similar dune behaviour is obtained as with the more complex k-$\varepsilon$ model of Shimizu et al. (2009) [14] for a flood wave corresponding to flume conditions.

Step lengths have been allowed to become larger than with the step length model of Shimizu et al. (2009) [14]. In addition, higher step lengths than the maximum observed in the experiments of Nakagawa and Tsujimoto (1980) [23] have been adopted such that effects of suspended load are implicitly taken into account. We would like to note that this implicit formulation is a first step to account for suspended sediment transport processes in a strongly schematized way, which needs to be extended and improved in further studies. It was shown that in this way, the model is able to predict a transition to upper-stage plane bed and clearly shows hysteresis effects due to the lag between bedform dimensions and discharge. While the simulated associated moments in time and the maximum dune height are close to the results of Shimizu et al. (2009) [14], the exact results in terms of water depth still showed differences of about 25%.

A scenario corresponding to the 1998 flood in the river Waal was modelled as well. While the qualitative behaviour of the dunes was represented well, the dune height was overestimated. This signifies the need for further model improvement/calibration. For the river scenario, the step length is significantly higher than the values used for the flume conditions. It was shown that the Waal model is also computationally fast. By slightly

increasing the discharge of the original river scenario, a transition to upper-stage plane bed was also modelled. Although in 1995, for much higher discharges, no transition to upper-stage plane bed has occurred (Wilbers and Ten Brinke (2003) [51]), this does show the sensitivity of the dune dynamics in this transitional regime and also indicates the need for a well-validated dune evolution model for extreme discharges.

It should be noted that when a transition to upper-stage plane bed indeed occurs during extreme floods (i.e., during dike design conditions), this has the potential to greatly reduce the design water depth because the hydraulic roughness of the main channel would be significantly lower. However, in a bifurcation river system such as the river Waal, balancing effects distribute this lowering of the water levels over both branches, if it would occur in one branch only [4]. For the model results presented here, the difference in water levels in one branch is approximately 0.5 m. Although in reality this effect would also be dampened due to the presence of floodplains, which are not accounted for here, this is still considered as a significant effect.

**Author Contributions:** Conceptualization, O.J.M.v.D., S.J.M.H.H. and J.S.R.; methodology, O.J.M.v.D., S.J.M.H.H. and J.S.R.; software, O.J.M.v.D.; resources, S.J.M.H.H.; writing—original draft preparation, O.J.M.v.D., S.J.M.H.H. and J.S.R.; writing—review and editing, O.J.M.v.D., S.J.M.H.H. and J.S.R.; visualization, O.J.M.v.D.; supervision, S.J.M.H.H. and J.S.R.; project administration, S.J.M.H.H.; funding acquisition, S.J.M.H.H. All authors have read and agreed to the published version of the manuscript.

**Funding:** This research was funded by NWO-AES, grant number 10483.

**Institutional Review Board Statement:** Not applicable.

**Informed Consent Statement:** Not applicable.

**Acknowledgments:** This study was carried out as part of the project 'BedFormFlood', supported by the Technology Foundation STW, the applied science division of now, and the technology programme of the Ministry of Economic Affairs. The authors thank Anke Wigger and Dominique van de Meché for their help in editing the manuscript.

**Conflicts of Interest:** The authors declare no conflict of interest.

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
