# Peer review of "Modelling Regime Changes of Dunes to Upper-Stage Plane Bed in Flumes and in Rivers"

_applsci, doi:10.3390/app112311212_

Round 1

Reviewer 1 Report

Manuscript title :  Modelling regime changes of dunes to upper-stage plane bed 2 in flumes and in rivers

The work describes  a model aimed to simulate the morphodynamic  evolution  of  dunes in the transition  to the upper plane bed by introducing  an alternative  sediment transport  model  in the  Paarlberg et al, model (2009).  The proposed model has been tested against  laboratory  and field data from literature.

The problem addressed by the authors is of interest  from a theoretical and  practical point of view. A correct prediction of  the rating curve during flood event is undoubtedly  an important contribution  to the hydraulic risk  evaluation.

The manuscript  can be improved  in  clarity  since  in several parts many concepts  are not  well explained. Some major comments  have to be made, as follows:

In the new step model the  effect  induced by suspension  seems to be taken in account  by introducing the  water depth (eq. 19)  However, the  adopted relationship  seems to include only the non dimensional  grain shear stress (Fig. 5).  In which way  is suspension   affecting  step length ?

Fig. 5 – it is not clear which equation has been used to plot the proposed step length model. 

L 489  in the dune regime,  bed load  is related to the effective  (or grain)  shear stress  while suspended load  is  mainly related to the total shear stress.  At the transition,  grain shear stress and total shear  stress  tend to be the same while  total sediment transport is  increasing,  At this stage  is very difficult to predict  the  fraction  of suspended  and  bed load . It is  reasonable to expect  a non linear  behaviour  of the step length  in contrast with  which that  shown  in Fig. 5.

Fig, 8 – which equation  is  used to  plot  the non-dimensional step length ? Could you specify  the range of the step length for which dunes are present ?

Fig. 9 – It  seems  that dunes are present for  specific discharge  in the range   0.1- 0.15 m2/s.  Outside this range  lower and upper plane bed should be present.  In particular,  in lower plane bed, which is a non natural  condition,  sediment transport  is almost  zero.  How can it be possible to have  dunes  in this range  during the falling  stage  of the flood, as shown in this figure ?

L 607  Since the  parameter  α  is non dimensional, how can the scale effect be explained  ?

The results obtained  by the proposed model  can be compared to the existing criteria for bed form regime (e.g. Vanoni, 1974) where extended experimental results have been used. Of particular interest could be  a  plot  of the model results in terms of Froude number in order to see at which values the transition and the extension of the upper plan bed is occurring.

Some minor comments  are  reported in the yellow notes  in the attached pdf file.

Reviewer 2 Report

The authors propose a morphological model able to predict the evolution of dunes for a broad range of flow conditions. They propose that the step length should be included as a function of the bed shear stress associated to the grain and the water depth. The model is able to capture the transition from dunes to the upper-stage plane bed and bedform hysteresis, as seen in Section 5 and in Section 6.

The manuscript is essentially well structured. English is good but there are some colloquialisms and non-technical style that should be corrected. Hence, some restructuring and re-writing would improve readability. Examples:

- consider merging 2.2 and 2.3 (there is no relevant information in 2.2);

- consider merging section 2.5 in section 3;

- improve caption in Figure 10;

- improve English in “albeit for the time being in a strongly schematized way” (or simply remove the colloquialisms) (L465)

- improve English in “still very computationally efficient” (L800).

There was an error rendering the references – there are numerous “Error! Reference source not found” throughout the text, which makes reading rather difficult.

There are some language imprecisions that should also be corrected. Examples:

- “step length increase with increasing flow strength” (L12, L19, L118) – please define flow strength or use more appropriate terms;

- the description of suspended sediment and the reference to “turbulent vortices” (L393);

- “turbulent eddy viscosity” (adjective turbulent is not needed (L399)

- the influence of turbulent diffusion is not at all clear in equation 17 and in parameter a; please clarify;

Although the model does seem to work, one of the premises of the work – the use of a bedload conceptual framework to describe suspended transport phenomena – should be better explained and justified. It appears that the major novelty introduced by the paper is equation 19. The parameter \alpha is the ratio of the step length and the median sediment diameter. For high values of representative Froude number the suspended load is dominant. However, equation 19 is still applied to avoid introducing specific suspended load formulations. This idea is not introduced with sufficient detail in the Introduction. One reads “step length … that combines bed load and suspended load processes”(L129) and wonders if it sense to model suspended sediment processes with a step length idealised for bed load. Section 4 should also provide more physical insights on the modelled phenomena, justifying why the bedload model is able to perform well. Also, the discussion in Section 8 should be modified to address the theoretical caveats of the model.

Round 2

Reviewer 1 Report

Accepted in the revised form

Author Response

We thank this reviewer. As no points are mentioned, we give no response.

Reviewer 2 Report

The authors have introduced several changes that meet my earlier comments.

There is still one issue to solve. The authors extend the use of a formulation for bedload to account for some effects of suspended sediment. It apparently works. 

I would like to know, in the opinion of the authors, why it works, if the fundamental principle is not correct.

I do not really care if the authors want to introduce disclaimers like "it is a first step", etc (e.g. L768-770).

Clearly there is some fundamental truth in the formulation - I would like that the authors would make the effort to identify, or isolate, the crucial elements of the formulation that make it work even if the starting point (Einsteins' step length) does not make sense. For instance, could the crucial element be an explicit out-phase forcing, relatively to the bed shear stress, with the introduction of the normalized flow depth?

This should be core of the discussion, I think it is within  reach for the authors.       
